# Plasma concentration of neurofilament light chain protein decreases after switching from tenofovir disoproxil fumarate to tenofovir alafenamide fumarate

Linn Hermansson[1,2], Aylin Yilmaz[1,2], Richard W. Price[3], Staffan Nilsson[4],
Scott McCallister[5], Tariro Makadzange[5], Moupali Das[5], Henrik Zetterberg[6,7,8,9],
Kaj Blennow[5,6], Magnus Gisslen[1,2]*

1 Department of Infectious Diseases, Institute of Biomedicine, Sahlgrenska Academy, University of Gothenburg, Gothenburg, Sweden, 2 Region Västra Götaland, Sahlgrenska University Hospital, Department of Infectious Diseases, Gothenburg, Sweden, 3 Department of Neurology, University of California, San Francisco, United States of America, 4 Mathematical Sciences, Chalmers University of Technology, Gothenburg, Sweden, 5 Gilead Sciences Inc, Institute of Neuroscience and Physiology, Foster City, California, United States of America, 6 Department of Psychiatry and Neurochemistry, University of Gothenburg, Gothenburg, Sweden, 7 Clinical Neurochemistry Laboratory, Sahlgrenska University Hospital, Mölndal, Sweden, 8 Department of Neurodegenerative Disease, UCL Institute of Neurology, Queen Square, London, United Kingdom, 9 UK Dementia Research Institute, UCL, London, United Kingdom

* magnus.gisslen@gu.se

## Abstract

### Background

Because tenofovir alafenamide (TAF) leads to significantly lower plasma tenofovir concentrations than tenofovir disoproxil fumarate (TDF) and is a stronger substrate for P-glycoprotein (P-gp) than TDF, TAF could lead to decreased central nervous system (CNS) tenofovir exposure than TDF. We aimed to determine if switching from TDF to TAF increases the risk of neuronal injury, by quantifying plasma levels of neurofilament light protein (NfL), a sensitive marker of neuronal injury in HIV CNS infection.

### Methods

Plasma NfL concentration was measured at baseline, week 24, and week 84 in stored plasma samples from 416 participants (272 switching to elvitegravir (E)/cobicistat (C)/emtricitabine (F)/TAF and 144 continuing E/C/F/TDF) enrolled in the randomized, active-controlled, multicenter, open-label, noninferiority Gilead GS-US-292-0109 trial.

### Results

While plasma NfL levels in both groups were within the normal range, we found a small but significant decrease in the E/C/F/TAF arm after 84 weeks from a geometric mean of 9.3 to 8.8 pg/mL (5.4% decline, 95% CI 2.0–8.4, p = 0.002). This change was significantly different (p = 0.001) from that of the E/C/F/TDF arm, in which plasma NfL concentration changed from 9.7 pg/mL at baseline to 10.2 pg/mL at week 84 (5.8% increase, 95% CI -0.8–12.9,

**Data Availability Statement:** All relevant data are within the manuscript and its Supporting Information files.

**Funding:** This study was financed by grants from the Swedish state under the agreement between the Swedish government and the county councils, the ALF-agreement (ALFGBG-717531, ALFGBG-715986 and ALFGBG-720931), the Swedish Research Council (#2018-02532 and #2017-00915), the European Research Council (#681712), the Knut and Alice Wallenberg Foundation, the Torsten Söderberg Foundation, the National Institutes of Health (R01 NS094067), and Gilead Sciences (CO-SE-292-4080). Gilead Sciences provided support in the form of salaries for authors [SM, TM, and MD], but did not have any additional role in the study design, data collection and analysis, decision to publish, or preparation of the manuscript. The specific roles of these authors are articulated in the 'author contributions' section.

**Competing interests:** We have the following interests: This study was funded in part by Gilead Sciences (CO-SE-292-4080). SM, TM, and MD are employees of Gilead Sciences. HZ has served at scientific advisory boards for Roche Diagnostics, Wave, Samumed and CogRx, has given lectures in symposia sponsored by Alzecure and Biogen, and is a co-founder of Brain Biomarker Solutions in Gothenburg AB, a GU Ventures-based platform company at the University of Gothenburg (all outside submitted work). KB has served as a consultant or at advisory boards for Alector, Biogen, CogRx, Lilly, MagQu, Novartis and Roche Diagnostics, and is a co-founder of Brain Biomarker Solutions in Gothenburg AB, a GU Venture-based platform company at the University of Gothenburg, all unrelated to the work presented in this paper. There are no patents, products in development or marketed products to declare. This does not alter our adherence to all the PLOS ONE policies on sharing data and materials.

p = 0.085). This increase is in line with what could be expected in normal ageing. Plasma NfL concentrations significantly correlated with age. No correlation was found between plasma NfL and serum creatinine.

## Conclusions

We found no biomarker evidence of CNS injury when switching from TDF to TAF. It is unclear whether the small decrease in plasma NfL found after switch to TAF is of any clinical relevance, particularly with plasma NfL levels in both arms remaining within the limits found in HIV-negative controls. These results indicate that switching from TDF to TAF appears safe with regard to neuronal injury.

## Introduction

HIV-1 enters the central nervous system (CNS) shortly after transmission, initiating a chronic infection in brain macrophages and microglia accompanied by intrathecal immune activation [1, 2]. If left untreated this may eventuate in neuronal damage [3]. Antiretroviral treatment (ART) inhibits CNS HIV replication and decreases the intrathecal immune activation substantially, although not to fully normal levels [4, 5].

Neurofilament light protein (NfL) is a major structural protein of axons that is highly expressed in the cytoplasm of large myelinated axons [6]. NfL can be quantified in cerebrospinal fluid (CSF) and blood, and increased levels are detected in both of these fluids in a variety of different neurodegenerative diseases [7–12]. Additionally, CSF NfL has been shown to be a sensitive marker of neuronal injury in HIV infection [13, 14]. While highest levels of CSF NfL are found in patients with HIV-associated dementia (HAD) and opportunistic CNS infections, increased CSF NfL concentrations can also be detected in HIV-infected individuals with axonal injury without overt neuro-symptomatic disease, mainly in those with low $CD4^+$ T-cell counts [13, 15, 16]. ART significantly reduces CSF NfL concentrations to the normal range, though to levels slightly higher than those of HIV-negative controls matched to lifestyle factors [15, 17].

NfL concentrations in plasma are 50 to 100 times lower than in CSF, but a recently developed ultra-sensitive method has made it possible to quantify NfL also in plasma [18] and plasma NfL concentrations strongly correlates with those of CSF [18–22].

Emtricitabine/tenofovir disoproxil fumarate (F/TDF) has been one of the most widely-used combinations of nucleoside reverse transcriptase inhibitors (NRTIs) for many years. TDF is a prodrug that is converted to tenofovir in blood and subsequently to the active form, tenofovir diphosphate (DP), intracellularly. High plasma levels of tenofovir are required to reach sufficient intracellular levels of tenofovir DP. Tenofovir alafenamide fumarate (TAF) is a more recently registered prodrug that is taken up intracellularly without conversion, mainly in lymphocytes and macrophages, and thereafter metabolized to its active form. This leads to significantly lower plasma tenofovir levels and higher intracellular levels [23, 24] and also to a lower risk of renal and bone toxicity [25, 26].

Concerns have been raised regarding potentially reduced CNS exposure of tenofovir, when administered as TAF compared to TDF. Both are substrates for the transport protein P-glycoprotein (P-gp) which means that they are subject for active blood-brain barrier efflux. TDF, however, is rapidly converted to tenofovir in the systemic circulation and relatively high tenofovir concentrations are reached in the CSF. Tenofovir is not a substrate of P-gp [27–30]. On

the contrary, tenofovir concentrations are low in both plasma and CSF during TAF treatment [31].

The Gilead GS-US-292-0109 was a randomized, active-controlled, multicenter, open-label study in which HIV-1-infected adults on a regimen containing TDF were included. All participants were virologically suppressed and had an estimated glomerular filtration rate (eGFR) of 50 mL/min or higher. They had been on their regimen containing TDF/FTC along with either elvitegravir/cobicistat (E/C), efavirenz, cobicistat-boosted atazanavir, or ritonavir-boosted atazanavir for at least 96 weeks. Participants were randomized 2:1 to switch to the single tablet regimen E/C/F/TAF or to continue their TDF-containing regimen. Switching to E/C/F/TAF was shown to be non-inferior regarding virological suppression and had a beneficial effect on proximal renal tubular function and bone mineral density [25, 26].

In the present study, we included only the subgroup of participants in the GS-US-292-0109 trial who were on E/C/F/TDF at baseline. Thus, the only difference between the arms was whether they continued with TDF or switched to TAF, the rest of their combination stayed stable with E/C/F. Because of the potential low CNS tenofovir exposure with TAF-treatment, our aim was to investigate whether switching to TAF was associated with increased neuronal injury compared to continuing with TDF, as measured by plasma levels of NfL.

## Methods

### Study subjects

From the GS-US-292-0109 (ClinicalTrials.gov: NCT01815736) we retrospectively included HIV-1-infected adults ($\geq$ 18 years) on treatment with E/C/F/TDF at baseline who either continued E/C/F/TDF or switched to E/C/F/TAF. Patients with remaining stored plasma samples from baseline, week 24, and week 84 were included. While the randomized phase of the study continued up to 96 weeks, availability of stored plasma at week 96 was restricted. We therefore chose week 84 for long-term follow-up.

The study was approved by the U.S. FDA and by Institutional Review Boards at all study sites. All participants signed written informed consent.

### Measurements

Plasma NfL concentrations were measured using a previously described ultrasensitive ELISA on the Single molecule array platform (Simoa; Quanterix, Lexington, MA, USA) [18]. All plasma samples were analyzed once in a single run at the Clinical Neurochemistry Laboratory at the University of Gothenburg by board-certified laboratory technicians blind to clinical data. A single batch of reagents was utilized; intra-assay coefficients of variation were below 10%.

All other analyses, including plasma HIV RNA and serum creatinine, were performed within the GS-US-292-0109 trial [25].

### Statistical analysis

Continuous variables, with the exception of age, were $\log_{10}$ transformed to approximate the normal distribution and then back transformed to present geometric means on the original scale. Comparisons within the groups were performed with paired sample t-test, and comparisons between the groups used independent t-test. Correlations were determined with Pearson correlation. Statistical analysis was performed using SPSS statistics (IBM SPSS version 25) or Prism (GraphPad Software version 7.0).

## Results

Of the 1443 participants in the GS-US-292-0109 trial, 459 were on E/C/F/TDF at baseline and 416 met our inclusion criteria with stored samples at baseline, week 24, and week 84. Of these, 272 switched to E/C/F/TAF and 144 continued with E/C/F/TDF. All had plasma HIV RNA < 50 copies/mL at baseline and during follow-up. Median (IQR) age was 41 (33–48) years and 8.4% were women. Baseline demographics were balanced between the two treatment groups with the exception of ethnic origin; similar to the complete GS-US-292-0109 trial, more patients in the TAF arm than in the TDF arm reported Hispanic or Latino ethnic origin (Table 1). Demographics of the 43 patients who did not meet the inclusion criteria were not different from the 416 included (data not shown).

At baseline, there was no significant difference in plasma NfL concentrations between the two arms. From baseline to week 84 there was a small but statistically significant decrease in plasma NfL in the E/C/F/TAF arm from 9.3 to 8.8 pg/mL (5.4% decline, 95% CI 2.0–8.4, $p = 0.002$). The change was significantly different ($p = 0.001$) from the E/C/F/TDF arm, in which plasma NfL concentration changed from 9.7 to 10.2 pg/mL (5.8% increase, 95% CI -0.8–12.9, $p = 0.085$). This increase is in line with plasma NfL changes by normal ageing. Based on data from HIV-negative controls (18), plasma NfL could be expected to increase with 5.3% in 84 weeks. There were no significant changes in plasma NfL from baseline to week 24 in any of the groups (Fig 1). Plasma NfL was significantly correlated with age ($r = 0.44$, $p < 0.001$ at baseline) but not with gender or ethnicity.

Serum creatinine levels as well as estimated glomerular filtration rate by Cockcroft-Gault formula (eGFR) were similar in both groups at baseline. Creatinine decreased significantly between baseline and week 84 in both arms, from 1.14 to 1.08 mg/dL ($p<0.001$) in the TAF-arm, and from 1.19 to 1.17 mg/dL ($p = 0.01$) in the TDF-arm. The decrease was significantly larger in the TAF-arm compared to the TDF-arm ($p = 0.02$) and the levels were significantly lower in the TAF-arm at week 84 ($p = 0.006$) (Fig 2). eGFR increased from 102.8 to 109.8 mL/min ($p < 0.0001$) in the TAF-arm, while no significant change (100.6 to 101.5 mL/min) was found in the TDF-arm.

**Table 1. Baseline characteristics of the participants.**

|  | Tenofovir alafenamid group (n = 272) | Tenofovir disoproxil fumarate group (n = 144) | p-value |
|---|---|---|---|
| **Age (years)** | 40 (32–48) | 42 (33–49) |  |
| **Women** | 23 (8.5%) | 12 (8.3%) |  |
| **Race** |  |  |  |
| Native American | 1 (0.4%) | 0 |  |
| Asian | 10 (3.7%) | 5 (3.5%) |  |
| Black | 56 (20.6%) | 38 (26.4%) |  |
| Native Hawaiian | 3 (1.1%) | 0 |  |
| White | 189 (69.5%) | 99 (68.8%) |  |
| **Ethnic Origin** |  |  |  |
| Hispanic or Latino | 65 (23.9%) | 18 (12.5%) | 0.0065 |
| **Baseline body-mass index (kg/m$^2$)** | 25.8 (23.3–29.3) | 26.6 (23.5–29.1) |  |
| **CD4 count (cells per uL)** | 693 (536–848) | 683 (557–854) |  |
| **Serum creatinine (mg/dL)** | 1.14 (1.11–1.17) | 1.19 (1.14-1-25) |  |
| **Plasma NfL (pg/mL)** | 9.28 (8.81–9.78) | 9.66 (8.86–10.54) |  |

Serum creatinine and plasma NfL are geometric mean (95% confidence interval), all other data are median (IQR) or n (%).

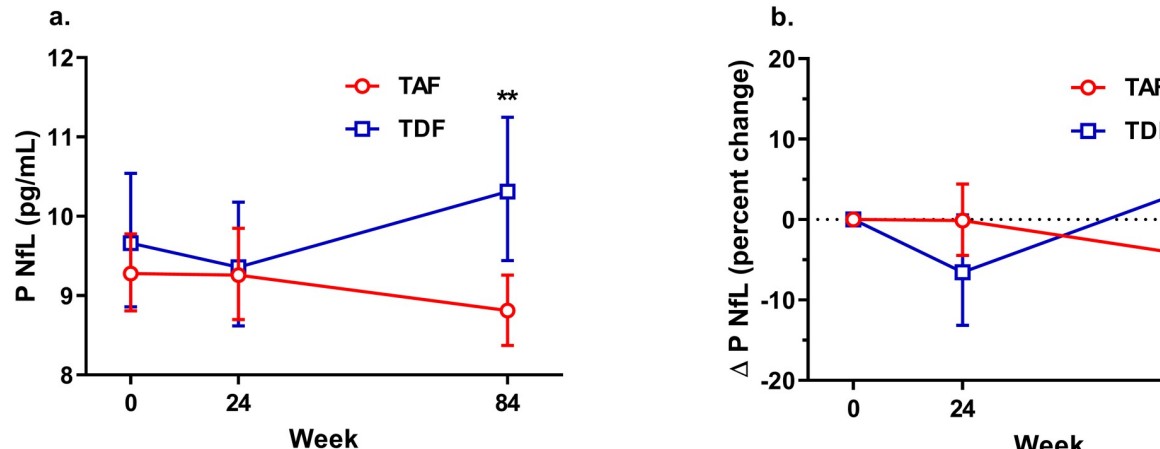

**Fig 1. Plasma NfL changes over time.** (a) Plasma NfL concentrations in participants on tenofovir alafenamide fumarate (TAF) (red, n = 272)) and tenofovir disoproxil fumarate (TDF) (blue, n = 144). Values are presented as geometric means and error bars indicate 95% confidence intervals. (b) Percent change in plasma NfL and 95% confidence intervals from baseline, to week 24 (TAF n = 271, TDF n = 142), and to week 84 (TAF n = 267, TDF n = 140). Plasma NfL was significantly higher in the TDF group and there was a significant difference between the groups in plasma NfL change from baseline to week 84.

There was no correlation between plasma NfL and serum creatinine at baseline (r = -0.07, p = 0.18) or at week 84 (r = -0.06, p = 0.25) (Fig 3). No significant correlation was found between changes in plasma NfL and serum creatinine between baseline and week 84 while there was a weak correlation between delta plasma NfL and delta eGFR (r = -0.013, p = 0.009).

## Discussion

Our hypothesis prior to initiation of the study was that switching from TDF to TAF could pose a risk of neuronal injury due to decreased CNS exposure of tenofovir, leading to a measurable increase in plasma NfL concentrations. Unexpectedly, we found the opposite: a small, but

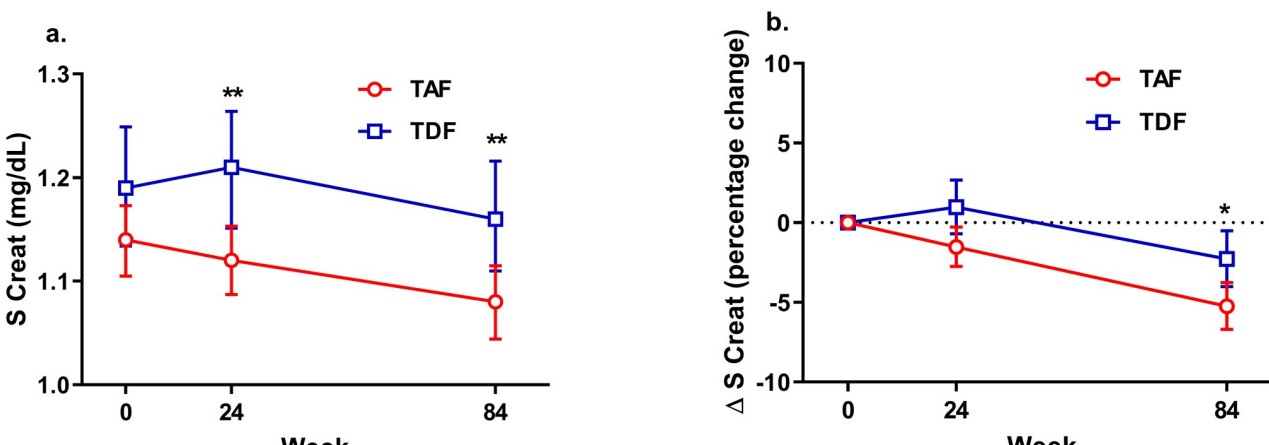

**Fig 2. Serum creatinine changes over time.** (a) Serum creatinine in participants on tenofovir alafenamide fumarate (TAF) (red, n = 272) and tenofovir disoproxil fumarate (TDF) (blue, n = 142). Values are presented as geometric means and error bars indicate 95% confidence intervals. (b) Percent change in serum creatinine and 95% confidence intervals from baseline, to week 24 (TAF n = 271, TDF n = 141), and to week 84 (TAF n = 265, TDF n = 138). Creatinine was significantly higher in the TDF group at week 24 and 84 and there was a significant difference between the groups in serum creatinine change from baseline to week 84.

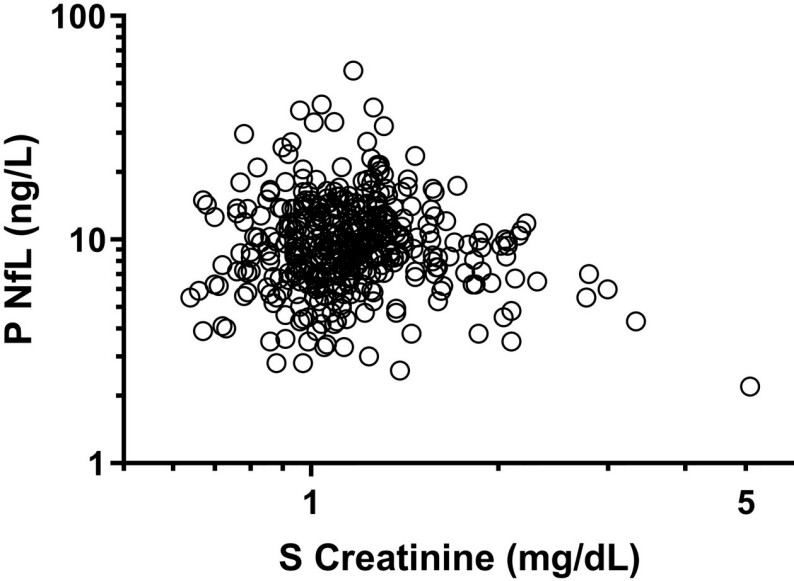

**Fig 3. No association between serum creatinine and plasma NfL.** No significant correlation between plasma NfL and serum creatinine at baseline (r = -0.07, p = 0.18). TAF: tenofovir alafenamide fumarate; TDF: tenofovir disoproxil fumarate; NfL: neurofilament light protein.

statistically significant, decrease in plasma NfL in the group receiving TAF 84 weeks after the switch. CSF tenofovir concentrations are six times lower when administered as TAF compared to TDF [31, 32], but since CSF concentrations do not correlate with intracellular concentrations [33, 34], the levels in macrophages and microglia were most likely high enough to have sufficient antiretroviral effect and prevent an increase in HIV-related CNS injury.

The mechanisms behind the reduction in plasma NfL 84 weeks after changing from TDF to TAF are unclear. One possible explanation could be the increased intracellular concentrations of tenofovir DP in macrophages and possibly microglia with TAF as compared to TDF, which may contribute to reduced neuronal injury through better virological suppression [24, 35]. There is, however, no evidence of insufficient inhibition of CNS viral replication in patients on ART and suppressed plasma and CSF viral load, and treatment intensification does not decrease CNS viral replication or immune activation [36, 37].

A second possible explanation could be that high plasma tenofovir concentrations associated with TDF might be more toxic to neurons than the lower concentrations from TAF treatment, and that this could increase plasma NfL levels. To our knowledge, however, there are no data regarding potential harmful effects of TDF on either CNS or peripheral neurons. TDF was not associated with neurotoxicity in a study investigating the toxicity of different antiretrovirals on rat brain cell cultures [38], nor has it been associated with a high rate of neuropathy. In addition, plasma NfL concentrations did not increase significantly during the 84-week long study period in the TDF arm, suggesting, at least, no progressive neuronal injury.

Previous studies have shown that there are correlations between cognitive decline and renal impairment, but there are no studies on renal impairment and NfL levels [39, 40]. TAF is associated with less tubular side effects compared to TDF, especially when administered together with ritonavir or cobicistat as in this study. As a third possibility for the reduction of plasma NfL that we considered was that the improved tubular function with TAF might increase clearance of NfL or substances harmful to neurons. There was, however, no association between creatinine and NfL at baseline or follow up, speaking against increased elimination of NfL due

to improved renal function in patients switching to TAF. It is unclear whether the small decrease in plasma NfL found after switch to TAF is of any clinical relevance, particularly with plasma NfL levels in both arms remaining within the limits found in HIV-negative controls.

Major strengths with this study are the large number of participants, the longitudinal design with randomization, and long follow-up. One important weakness is that we did not have a HIV-negative control group. The normal range of plasma NfL has not been fully determined, but the levels we found are in the same range as has been found in healthy controls in other studies [18, 41]. Another weakness is that no CSF samples were available making it impossible to analyse if changes in NfL were associated with changes in CNS immune activation or other CSF biomarkers that might have elucidated underlying mechanisms.

Furthermore, we cannot exclude a potential informative censoring and consequent attrition bias with differences between the included 416 patients and the 43 who did not meet the inclusion criteria, even if they had similar demographics at baseline.

## Conclusions

In summary, we found that plasma NfL decreased significantly 84 weeks after switching from E/C/F/TDF to E/C/F/TAF. The clinical significance of, and the mechanisms behind, this decrease within the normal range, are unclear. Nonetheless, switching from TDF to TAF appears safe with regard to neuronal injury.

## Supporting information

**S1 File. Complete dataset.** Included in the supporting information is an excel file with the complete dataset used for analysis.
(XLSX)

## Author Contributions

**Conceptualization:** Richard W. Price, Scott McCallister, Magnus Gisslen.

**Data curation:** Scott McCallister, Tariro Makadzange, Moupali Das.

**Formal analysis:** Linn Hermansson, Aylin Yilmaz, Richard W. Price, Staffan Nilsson, Henrik Zetterberg, Kaj Blennow, Magnus Gisslen.

**Funding acquisition:** Henrik Zetterberg, Magnus Gisslen.

**Methodology:** Aylin Yilmaz, Richard W. Price, Staffan Nilsson, Henrik Zetterberg, Magnus Gisslen.

**Project administration:** Magnus Gisslen.

**Resources:** Kaj Blennow, Magnus Gisslen.

**Supervision:** Magnus Gisslen.

**Validation:** Henrik Zetterberg.

**Writing – original draft:** Linn Hermansson.

**Writing – review & editing:** Linn Hermansson, Aylin Yilmaz, Richard W. Price, Staffan Nilsson, Scott McCallister, Tariro Makadzange, Moupali Das, Henrik Zetterberg, Kaj Blennow, Magnus Gisslen.

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
