## [Decision Letter · Decision Letter 0]

25 Sep 2019

PONE-D-19-22861

Plasma concentration of neurofilament light chain protein decreases after switching from tenofovir disoproxil fumarate to tenofovir alafenamide fumarate

PLOS ONE

Dear Prof. Gisslen,

Thank you for submitting your manuscript to PLOS ONE. After careful consideration, we feel that it has merit but does not fully meet PLOS ONE’s publication criteria as it currently stands. Therefore, we invite you to submit a revised version of the manuscript that addresses the points raised during the review process.

We would appreciate receiving your revised manuscript by Nov 09 2019 11:59PM. To enhance the reproducibility of your results, we recommend that if applicable you deposit your laboratory protocols in protocols.io, where a protocol can be assigned its own identifier (DOI) such that it can be cited independently in the future. For instructions see: http://journals.plos.org/plosone/s/submission-guidelines#loc-laboratory-protocols

We look forward to receiving your revised manuscript.

Kind regards,

Alan Winston

Academic Editor

PLOS ONE

Journal Requirements:

2. Please include the full name of the IRB that approved your study in the ethics statement.

3. Please provide additional details regarding participant consent. In the Methods section, please ensure that you have specified (1) whether consent was informed and (2) what type you obtained (for instance, written or verbal). If your study included minors, state whether you obtained consent from parents or guardians. If the need for consent was waived by the ethics committee, please include this information.

4. Please include the registration number for the clinical trial referenced in the manuscript.

6. Thank you for stating the following in the Competing Interests section:

'HZ has served at scientific advisory boards for Roche Diagnostics, Wave, Samumed

and CogRx, has given lectures in symposia sponsored by Alzecure and Biogen, and is

a co-founder of Brain Biomarker Solutions in Gothenburg AB, a GU Ventures-based

platform company at the University of Gothenburg (all outside submitted work).

KB has served as a consultant or at advisory boards for Alector, Biogen, CogRx, Lilly,

MagQu, Novartis and Roche Diagnostics, and is a co-founder of Brain Biomarker

Solutions in Gothenburg AB, a GU Venture-based platform company at the University

of Gothenburg, all unrelated to the work presented in this paper.

SM, TM, and MD are employees of Gilead Sciences.

The other authors declare no competing interests.'

We note that one or more of the authors are employed by a commercial company: Gilead Sciences.

Additional Editor Comments (if provided):

Reviewers' comments:

Reviewer's Responses to Questions

**Comments to the Author**

1. Is the manuscript technically sound, and do the data support the conclusions?

Reviewer #1: Partly

Reviewer #2: Yes

Reviewer #3: Yes

Reviewer #4: Yes

2. Has the statistical analysis been performed appropriately and rigorously? 

Reviewer #1: Yes

Reviewer #2: No

Reviewer #3: Yes

Reviewer #4: Yes

3. Have the authors made all data underlying the findings in their manuscript fully available?

Reviewer #1: Yes

Reviewer #2: Yes

Reviewer #3: Yes

Reviewer #4: Yes

4. Is the manuscript presented in an intelligible fashion and written in standard English?

Reviewer #1: Yes

Reviewer #2: Yes

Reviewer #3: Yes

Reviewer #4: Yes

5. Review Comments to the Author

Reviewer #1: Authors report on a surrogate maker, a neurofilament protein, in the plasma to report on neurotoxicity induced by HIV replication. They compare neurotoxicity under tenofovir treatment with toxicity under TAF treatment. Analysis is well done but conclusions should be downplayed due to the lack of a negative control group, different sizes in the groups affecting the significance level , lack of measurement in CSF or orthogonal measurement of HIV replication versus drug induced neurotoxicity. The conclusion that switching to TAF does not increase neurotoxicity compared to tenofovir treatment, using this indirect method, can be made but the discussion and conclusions on the so-called significant decrease in the filament level is not substantiated. It seems that plasma levels were normal and remain normal. Conclusions should be rewritten and re-discussed. Do authors plan to use other methods to investigate HIV replication in the brain in more detail ?

Reviewer #2: This a post-hoc analysis of the Gilead GS-US-292-0109 trial restricted to people who were on E/C/F/TDF. This has been done so that the only difference between the arms would be using TAF instead of TDF after switching with the rest of the drugs in the regimen being identical. Thus, a cleaner dataset in which any difference in markers after switching could be attributed to TAF vs. TDF. Because, people originally enrolled in the trial were at time zero randomized either to stay on TDF or switch to TAF, exchangeability is also retained in this subset of data which was included for analysis. The aim was to investigate whether switching to TAF was associated with increased neuronal injury compared to remaining on TDF. The manuscript is clearly written but there are few main points that need to be addressed:

1. Although groups were exchangeable at baseline, out of the 459 who were on E/C/F/TDF at baseline, only 414 had week 84 samples/values so there is potential attrition bias in this analysis. It is crucial to address this issue in more details than what was done by the authors in the present form:

i) Table 1 should show all potential common causes of treatment allocation and risk of neuronal injury. Ideally such a table should be identical to that shown in the parental paper published in Lancet Infectious Diseases [Lancet Infect Dis. 2016 Jan;16(1):43-52]. This is to show that attrition did not create imbalances in important potential confounders.

ii) If imbalances in important confounders are detected, multivariable analysis should be conducted to control for these. Ideally, this analysis should be performed using marginal methods evaluating the effect of treatment after averaging out the effect of the covariates. This is because authors are interested in the causal effect of switching to TAF if there was no attrition bias. Specifically, marginal weighted linear regressions models should be employed.

iii) Attrition bias should be clearly reported in the Figures by adding a footonote with the number of people contributing measurements at the 24 and 84 week time points.

iv) The issue of potential informative censoring and consequent attrition bias (as people who are retained in TAF up to week 84 are likely to be those who have less injury) needs also to be discussed as a limitation.

v) Authors should explain why numbers in Table and in the text are inconsistent (416 total vs 414 in the text and abstract, 144 in the TDF arm vs. 142 in the text)

vi) Authors should also discuss power and potential issues related to the chosen target population in the trial. Indeed, contrary to what shown in the parental paper, there was no improvement in renal function in the TAF arm in this subset. This shades doubt also on the comparison of the risk of neuronal injury.

vii) Clinical relevance of the small difference found is rightly questioned in the abstract. A discussion of this point should be included also in the main text.

Reviewer #3: The hipothesis of your study is interesting and your conclusion supported the idea that TAF was a drugs with lower toxicity than TDF.

Some little revision are however due before publish your paper

In tab 1 you present data registered at baseline. Could be better to show in tab any difference within the two group also regarding age creatinine and nfl showing the p value.

It is possible that the different number of patients on TAF anf TDF (2:1) could modify the statistical significance of your results: really the difference between Nfl or creatinine could be modified by study designe and patients distribution into the the two arms. Please discuss about any possible bias linked to this aspect.

At the same time the value of creatinine could be not perfectly associated to the real filtration capacity of every patient. A GFR calculation could give us a more exact estimation of renal function.

Finally the figures showed a different trend of Nfl concentrations at 24 and 84 weeks, mainly in TDF group. Could you speculate about this different bifasic dynamic?

Reviewer #4: This is a well-thought through and well-written paper, on an interesting and clinically relevant hypothesis.

1. Line 35: for 10.3 to 9.6 pg/mL: are these the median values? or mean? Need to state. The values presented here are different from the values presented in the main text Results section(9.3 to 8.8 pg/mL): was one set of values: median, and one set of values: mean?

Line 37: Once again the reported change in plasma NfL in the abstract (11.1 to 11.7pg/mL) is different from what is reported in the main test results (9.7 to 10.2pg/mL). When you say 'at follow up', do you mean week 84?

2. Line 36: Does the p<0.01 value relate to the comparison of the percentage change in both arms? If so, you should state that; this sentence is very confusing to read. Also worth stating the median/mean percentage change in both arms at week 84, to put that p-value into context.

3. Line 114: How many times were each sample analysed? Were the samples analysed in duplicate/triplicate?

4. Line 135: Table 1 does not show the p-values for statistically significant differences in age and gender, so the reference to Table 1 should be shifted to the sentence before. Referencing Table 1 here implies you are demonstrating p-values for differences in age and gender in Table 1.

Line 142: What is the p-value for difference in plasma NfL between the 2 arms at baseline? Since you describe the p-values, you should put it in Table 1.

5: Line 148: What other factors did you look at for associations with plasma NfL? Other factors related to eGFR perhaps: gender, ethnicity, weight?

6: Line 219: to also add the lack of age-related reference values for plasma NfL- unlike CSF NFL which has age-related reference values, and thus may be a more reliable/ interpretable biomarker.

7: The lack of concurrent CSF sample needs to be acknowledged in the discussion, given that plasma NfL using Simoa is still novel, and interpretation is more difficult for the reasons mentioned above

8: I agree with the final conclusions drawn from this paper.

6. PLOS authors have the option to publish the peer review history of their article (what does this mean?). If published, this will include your full peer review and any attached files.

Reviewer #1: No

Reviewer #2: No

Reviewer #3: No

Reviewer #4: No

---

## [Editor Report · Decision Letter 1]

25 Nov 2019

Plasma concentration of neurofilament light chain protein decreases after switching from tenofovir disoproxil fumarate to tenofovir alafenamide fumarate

PONE-D-19-22861R1

Dear Dr. Gisslen,

We are pleased to inform you that your manuscript has been judged scientifically suitable for publication and will be formally accepted for publication once it complies with all outstanding technical requirements.

With kind regards,

Alan Winston

Academic Editor

PLOS ONE
---

## [Editor Report · Acceptance letter]

3 Dec 2019

PONE-D-19-22861R1 

Plasma concentration of neurofilament light chain protein decreases after switching from tenofovir disoproxil fumarate to tenofovir alafenamide fumarate 

Dear Dr. Gisslen:

I am pleased to inform you that your manuscript has been deemed suitable for publication in PLOS ONE. Congratulations! Your manuscript is now with our production department. 

With kind regards,

on behalf of

Prof. Alan Winston 

Academic Editor

PLOS ONE